# The Global Pandemic as a Life-Changer? Medical, Psychological, or Self Help during COVID-19 Pandemic: A Cross-Sectional Representative Study

**DOI:** 10.3390/ijerph20021092

**Published:** 2023-01-07

**Authors:** Tomasz Sobierajski, Stanisław Surma, Monika Romańczyk, Marek Krzystanek

**Affiliations:** 1Faculty of Applied Social Sciences and Resocialization, University of Warsaw, 26/28 Krakowskie Przedmieście Str., 00-927 Warsaw, Poland; 2Clinic of Psychiatric Rehabilitation, Faculty of Medical Sciences in Katowice, Medical University of Silesia, 45/47 Ziołowa Str., 40-635 Katowice, Poland

**Keywords:** antidepressants, psychiatry, public health, society, SARS-CoV-2

## Abstract

The survey was conducted on a representative adult sample of Poles one year after the announcement of the global COVID-19 pandemic. The survey aimed to determine how the public in different social groups and age categories assessed the impact of the pandemic on their personal and professional lives, and where and to what extent respondents sought psychological and medical help to cope with the effects caused by the pandemic. The survey was conducted using the CAWI technique based on a questionnaire designed by an interdisciplinary team of experts. The study indicated that 61.9% of respondents declared that the COVID-19 pandemic did not bring any good, and had rather adverse effects on their lives, and 57.7% of respondents declared that the pandemic had not affected their professional lives. Nearly half of the respondents (45.0%) declared that although the pandemic forced them to change their personal lives, it did not work out for them. Due to the impact of the COVID-19 pandemic, every eighth respondent (12.3%) contacted a mental health specialist—a psychologist, or psychiatrist. Young people most often use psychological and medical help. Due to its representative nature, the survey can be used for in-depth qualitative analyses of the impact of the pandemic on people’s mental health.

## 1. Introduction

The COVID-19 pandemic had a very significant impact on many areas of life. The highly contagious SARS-CoV-2 virus—regardless of the subsequent variants that emerged—caused social and economic life to change, or even stop, as it did during the global lockdown in the spring of 2020. Furthermore, the effects of this global intervention are the subject of analysis [1,2].

The course of the pandemic was very dynamic. Initially, people were confined to their homes, and most countries worldwide restricted or wholly abandoned their ability to move internally and travel to other countries and continents. However, this did not reduce the development of the pandemic, although it slowed it down somewhat [3]. In mid-2020, more countries began to reduce the scope of restrictions. The policy of managing restrictions to protect public health varies, depending on the country, the epidemiological situation of the region, and the level of risk of contracting COVID-19. A year after the outbreak, and subsequent waves of coronavirus, people began to get used to living with the pandemic [4,5]. The big hope for resolving the pandemic crisis was the intensive development of vaccines against COVID-19. As of late December 2020, there was hope for resolving the pandemic in the European Union, thanks to the European Commission’s approval of BioNTech’s vaccine against COVID-19 to protect public health, and within a month, two more were approved [6]. Individual countries in Europe have introduced a vaccination schedule. In Poland, four stages of populational vaccination against COVID-19 have been introduced: stages zero, one, two, and three. Stage 0 covered mainly HWs. Stage 1 mainly covered people over 60. Stage 2 covered people under 60 with chronic diseases. Furthermore, from 12 April, a gradual process of including more HWs in the vaccination schedule began, which ended on 9 May [7]. 

Nevertheless, the first year of the COVID-19 pandemic was characterized by changing dynamics. These ranged from a global near-total lockdown and uncertainty about what the disease was and how dangerous it was to constant anxiety about when the next wave would come and what kind of restrictions it would entail. The constant threat of the virus, the restriction of social contact, forced or self-imposed isolation, and increases in mortality due to direct and indirect effects of the virus were not conducive to people’s mental well-being [8]. The COVID-19 pandemic—specifically, the fear associated with it and the restrictions put in place to protect public health—forced changes in many people’s personal and professional lives. These changes were associated with additional, often ongoing, stress. Some people sought specialized help to cope with the changes caused by the pandemic. During the global pandemic, stressors were not just related to the risk of infection. Sometimes people’s mental health was more threatened by elements such as job loss, lifestyle changes, and lack of, or reduced, social contact [9]. A meta-analysis by Brooks et al., based on papers describing people’s mental state during the quarantine period and published just before the announcement of the global lockdown caused by the SARS-CoV-2 virus, indicated that most reviews found adverse psychological effects in people [10]. Coifman et al. analyzed emotions and the resulting behaviors associated with the COVID-19 pandemic. They found that positive and negative emotions caused people to take prophylactic measures to prevent infection [11]. It confirms previous research while indicating that fear has a much more significant impact than happiness on taking part in pro-social behavior in a pandemic [12]. In the context of social behavior in mental health, it is worth noting that the risks associated with COVID-19 were complex for members of the public to assess because, especially during the first period of the pandemic, there was a lack of information regarding the short-term and long-term risks of SARS-CoV-2 infection.

Given the above, it is essential to conduct a study on a population-representative group of adults to answer several questions: What changes were people subjected to because of the COVID-19 pandemic?How did they assess the first year of the pandemic in terms of its impact on their professional and personal lives?Did they—and if so, to what extent—use specialists and pharmacology to cope with the stress the pandemic caused?

Finding answers to these questions in a multidisciplinary team of researchers seemed essential to assess the mental health of the public in a pandemic state and create proposals of patterns for public health policy action on a regional and national scale.

## 2. Materials and Methods

### 2.1. Study Design and Population

The survey was conducted one year after the WHO announced the pandemic [13]. A random sampling method was used to obtain the best socio-demographic representation of the subjects. The survey included 1000 adults, and each person was interviewed using a questionnaire prepared by the study authors. The survey was conducted on 30–31 March 2021, on a research panel using the CAWI (Computer Assisted Web Interview) technique. 

Thirty-eight million people were living in Poland at the time of the survey. The impact of the COVID-19 pandemic on Poland’s demographics was significant. The number of deaths in 2020 exceeded, by more than 100,000, the average annual value of the last 50 years (477,000 to 364,000), while the death rate per 100,000 inhabitants reached its highest value since 1951 [14]. The immediate cause of such an increase was the concordance due to COVID-19 or complications from COVID-19.

### 2.2. Questionnaire Design

The questionnaire used in this study was initially designed for this study, following the latest sociological, psychological, and methodological knowledge of the authors. The questionnaire consisted of 20 questions. Six of them were demographic and included information on age, year of birth, gender, place of residence, education, and income. The factual fourteen questions asked respondents to rate the change in their personal and professional lives due to the pandemic, their perceptions of life after the pandemic, and the impact of the pandemic on respondents’ self-esteem and emotions. Each question was closed. Answers to some of the questions were presented in a cafeteria.

Furthermore, for the remaining questions, an extensive Likert scale was used. The questionnaire was revised by qualified methodologists from the research company and then adapted to the technique used in the study. The questionnaire was extensively evaluated for the implementation of the study. For this purpose, a pilot study on a random group of 20 respondents was conducted to verify the correctness of the tool. After considering the methodological and technical comments, the questionnaire was technically adapted and included in the research panel of the research company, and then sent to the randomly selected respondents via a link.

### 2.3. Statistical Analysis

Descriptive statistics were conducted for presenting the demographic variables (gender, age, education, class of residence, and income). The relationship between variables was evaluated by using the Chi-squared test. Statistical analysis was performed in IBM SPSS Statistics 27.0.1.0 (IBM, Armonk, NY, USA). Answers to questions are presented with a total number of respondents (n) and frequencies of subgroup (%). For all analyzes, a *p*-level of < 0.05 was considered statistically significant.

### 2.4. Ethical Considerations

The SW Research Company that carried out the research is a member of ESOMAR (European Society for Opinion and Marketing Research) and provides the approval for ethical implementation of the research and the protection of respondents’ data. The quality of the tests and compliance of the test procedures with the standards is confirmed by the PKJPA (quality control program for researchers’ work) quality certificate granted to SW Research in 2015. Participants were informed about the purpose of the study and gave informed oral consent to participate in the study. When conducting social research that is carried out by a research company that operates under international ethical standards, the approval of the university ethics committee is not required. This research was a social survey carried out by a research company operating under international ethical standards. No sensitive data of the respondents’ is collected during this type of research. In such situations, the approval of the university ethics committee is not required.

## 3. Results

### 3.1. Participants Characteristics

The study was conducted on a representative sample of 1000 adult Poles. The detailed demographic distribution of the studied group, corresponding to the quota distribution of demographic variables in individual categories, is presented in Table 1.

### 3.2. Impacts of the Pandemic

According to most respondents, the COVID-19 pandemic did not bring people any good and had rather negative effects—61.9% of respondents. The remaining people (38.1%) answered that the COVID-19 pandemic forced some people to make positive changes and had relatively positive effects. When broken down by gender, assessments of the pandemic’s effects were similar (Table 2). Positive effects of the pandemic were slightly more likely to be cited by those between 35–49 years old (Table 3), those with primary education (Table 4), those from the largest cities (Table 5), and those who earn within the national average (Table 6). 

### 3.3. Impact of Pandemic on Life Changes

One in nine people surveyed (11.4%) declared that the COVID-19 pandemic forced them to change their professional life, but it worked out for them. Three out of ten people surveyed (30.9%) declared that they had to change their work life due to the pandemic, but it did not work out for them. The remainder—more than half of the respondents (57.7%)—declared that the pandemic had not affected their professional lives. The COVID-19 pandemic affected the professional lives of the women and men surveyed similarly (Table 7). Retirees were least affected by the pandemic in their professional lives (Table 8). The youngest respondents (Table 9), those from the largest cities (Table 10), and the highest earners (Table 11) were the most satisfied with the occupational changes caused by the pandemic.

One in nine respondents (11.5%) declared that the COVID-19 pandemic forced them to change their personal lives, but it worked out for them. Nearly half of the respondents (45.0%), declared that although the pandemic forced them to change their personal lives, it did not work out for them. The pandemic did not change their personal lives for 43.5% of respondents. The positive effects of changes in personal life caused by the COVID-19 pandemic were felt more often by men than women (Table 12), the youngest people under 24 (Table 13), those with primary education (Table 14), those from the largest cities and living in rural areas (Table 15), and the highest earners (Table 16).

A negligible number of respondents (2.9%) assessed that their lives had changed for the better due to the COVID-19 pandemic. One in ten people (10.4%) assessed that it had changed for the better. One in two people (48.5%) assessed that it had changed for the worse, and 16% assessed that it had definitely changed for the worse. One in five people (22.2%) declared that their lives had not changed at all during the pandemic. Life during the COVID-19 pandemic changed (for better or worse) more often for women than men. One in four men (26.1%) and one in five women (18.9%) reported that their lives had not changed at all during the pandemic. As a result of the pandemic, life changes were rated as better more often by men than women (11% for women vs. 15.8% for men—the sum of “changed definitely for the better” and “changed rather for the better” ratings) (*p* = 0.003) (Table 17). 

The COVID-19 pandemic had the most negligible impact on the lives of the youngest respondents under 24. In this group, 17.6% declared that nothing had changed in their lives because of it. In the other age groups, an average of one in four and one in five declared that the pandemic had not caused any changes in their lives. The higher the age, the more negatively assessed the impact of the COVID-19 pandemic on their lives, with 60% of 18–24 year-olds, 61.9% of 25–34 year-olds, 61.9% of 35–49 year-olds, 66.7% of 50–64 year-olds, and 71.1% of 65+ year-olds rating the life changes caused by COVID-19 as negative (sum of “changed rather worse” and “definitely changed worse” responses) (*p* = 0.001) (Table 17). 

Life changes caused by the COVID-19 pandemic definitely for the better were significantly more often declared by respondents with the primary education—every sixth person in this group (15.6%). As education increased, the percentage of respondents who did not feel any changes in their lives due to the pandemic increased significantly. At the same time, the percentage of those who believe that their lives have changed for the worse due to the COVID-19 pandemic increased with increasing education—50% of those with primary education, 61.5% of those with vocational education, 65% of those with secondary education, and 69.9% of those with higher education (sum of responses “changed rather for the worse” and “definitely changed for the worse”) (*p* = 0.004) (Table 17). 

According to three in ten respondents from a city between 200,000 and 499,000 residents (29.9%), one in four rural residents (25.4%), one in five residents of the largest cities with over 500,000 residents (19.7%), a city between 20,000 and 99,000 residents (20.1%), the smallest cities up to 20,000 residents (18.8%), and one in eight residents of a city between 100,000 and 199,000 residents, the COVID-19 pandemic has changed nothing in their lives. The larger the city, the worse the changes in respondents’ lives caused by the pandemic. One in five residents of cities with a population of 100,000 or more indicated the change as definitely for the worse (Table 17). 

On average, two-thirds of people in all income categories rate their lives as having changed rather for the worse, or definitely for the worse, because of the pandemic. The exception were respondents with the highest incomes. In this group, one in two people (51.5%) assessed that their lives had worsened because of the pandemic (Table 17).

### 3.4. Use of Specialist Assistance and Pharmacological Support during the Pandemic

Due to the impact of the COVID-19 pandemic, every eighth respondent (12.3%) contacted a mental health specialist—a psychologist or psychiatrist, of which 4.9% declared that it helped them a lot, and 7.4% said that it did not help them at all. Every third respondent (35%) did not use the help of this type of specialist, but did not exclude that they will use this help in the future. Furthermore, half of the respondents (52.8%) declare they do not need this help due to the pandemic.

Due to the pandemic, every eighth respondent (12.8%) contacted a psychiatrist, of which 3.6% declared it helped them, and 9.2% said it did not help them. Three out of ten respondents (29.7%) did not use support from a psychiatrist because of the pandemic but did not exclude that they will do so in the future, and 57.5% declared that the pandemic did not require this type of help for them.

In the group with the lowest level of education (primary), 22.4% of the respondents benefited from the assistance of psychiatrists (in the remaining education groups it was about 10%) and 16% from the assistance of psychotherapists (in the remaining groups, about 8%). This demographic group also stands out regarding the effectiveness of help from specialists. The response “Yes, it helped me a lot” concerning psychologists was indicated by 15.4% (in the remaining groups on average 5%), and in the case of psychiatrists by 9.1% (in the remaining groups, on average 3%).

Due to the pandemic, 13.1% of respondents began to lack antidepressants and drugs/medications to improve their mood, of which 5.1% say that it helped them a lot and 8% declare that it did not help them. Every fourth respondent (26.1%) claimed that due to the pandemic, they have not started using medications to improve their mood, but they do not exclude it in the future. Six out of ten respondents (60.9%) declared that due to the pandemic, they do not need to take medications that improve their mood (Table 18, Table 19, Table 20, Table 21 and Table 22).

### 3.5. Self-Assessment of the Mental State

Only 3.5% of respondents assessed their current mental state as “definitely optimistic”. Every fourth respondent (26.4%) assessed it as “rather optimistic”, every fifth respondent (19.6%) assessed it as “rather pessimistic”, and every twentieth (5%) as “definitely pessimistic”. The remaining respondents—45.4%, were not able to assess their condition, claiming that they are “neither pessimistic nor optimistic”.

Men were more optimistic than women (respectively: 35.6% vs. 25%—the sum of responses “definitely optimistic” and “rather optimistic”; *p* = 0.003), as well as young people up to 34 years of age and the oldest people over 65 (34% up to 24 years old), 33.4% between 25 and 34 years old, 25.8% between 35 and 49 years old, 28.2% between 50 and 64 years old, and 33% over 65—the sum of responses “definitely optimistic” and “rather optimistic”). Almost every second person (47%) with primary education declared that their mental state is optimistic. Every fourth person (25.6%) with vocational education, three out of ten people with secondary education (29.6%), and higher education (30.1%) declared that their mental state is optimistic. Regarding the place of residence, the least optimistic were the inhabitants of medium-sized cities with 200–499,000 inhabitants (22.7%). In the remaining places of residence, an average of 3 out of 10 respondents assessed their condition with optimism. People with the highest income above 5000 PLN (40.2%) were more optimistic than those with the lowest income (25.7%). Although, among people with the highest incomes, every fifth (22.8%) assessed their condition as pessimistic.

There is a significant statistical relationship between the self-assessment of mental state and the use of mental health specialists (psychotherapists/psychologists and psychiatrists) (Table 23).

### 3.6. Life after the Pandemic

According to most respondents, life will change after the COVID-19 pandemic. Nearly three in ten people believed (28.7%) that life—compared to life before the pandemic—would change in all areas, and one in two respondents (52.9%) believed that it would change, but only in some areas. The remaining respondents (18.4%) believe it will not change at all. Women were slightly more likely than men to agree that life after the pandemic will change in all areas (Figure 1), along with the youngest people under 24 and the oldest people—65 years and older (*p* = 0.006) (Figure 2), respondents with primary education (*p* = 0.005) (Figure 3), those who live in small and medium-sized cities (Figure 4), and those who earn up to 3000 PLN a month (Figure 5).

### 3.7. Nostalgia for the Past

The vast majority (92.7%) of respondents would like a return to their pre-pandemic COVID-19 lives, with two-thirds of respondents (67.2%) wanting a return to their pre-pandemic lives in all areas. One in four respondents (25.5%) wanted a return to their pre-pandemic lives in only some areas. The remaining people (7.3%) would not want a return to their pre-pandemic lives. 

Significantly more women than men (*p* < 0.001) would like to return to their life before the pandemic. In all areas (Figure 6), the oldest respondents were 65 years and older (Figure 7), those with vocational education (Figure 8), those earning between PLN 1001 and 3000 (*p* < 0.001) (Figure 9), and those living in small towns with up to 99,000 residents (Figure 10).

## 4. Discussion

The study presents the impact of the COVID-19 pandemic on the professional and personal lives of individuals in a cross-section of society, considering demographic criteria and the role and quality of psychological and medical support in the process of coping with the effects of the pandemic. In our study, we showed a significant relationship between pandemic-induced change in work life and personal life and the age, education, and income of the respondents. 

Our study was conducted one year after the outbreak of the global pandemic. It was a watershed moment in the pandemic as large-scale COVID-19 vaccination programs began to operate in developed countries, including Poland. However, as the analysis by Pandey et al. indicates, the introduction of population-based vaccination against COVID-19 did not equate to a reduction in pandemic-related stress, as the already known stress predictors of isolation and fear of contracting the disease were joined by stress related to fear of vaccination and post-vaccination side effects [15]. As research indicates, pandemics of this type causes people to first experience surprise, then uncertainty, which in some people can turn into permanent anxiety, and in some people adaptation processes will be triggered, in this case to a pandemic, uncertain situation [16,17,18,19]. Many studies confirm that the COVID-19 pandemic negatively affected respondents’ well-being in their professional and personal lives. Mainly due to social isolation [20,21,22,23]. However, our cross-sectional study indicates that the pandemic may also play the role of eustress, stimulating adaptive abilities. A Gibbons study that examined the impact of the COVID-19 pandemic on English students’ attitudes indicated that optimism was as practical as defensive pessimism in stimulating learning motivation [24]. In addition to individual predispositions, the impact on the type of response to a pandemic may be related to the time and intensity of experiencing distress. It is because stress is not a simple reaction. It is an interaction between the individual and the environment. An individual’s social and family situation, cognitive predisposition, life experience, gender, age, and level of education affect how an individual will respond to stress [25]. 

The subsequent results of this study confirm the possibility of an ambiguously negative impact of the pandemic on functioning. As older respondents, the percentage of those who felt that the changes in their professional lives caused by the COVID-19 pandemic worked out for them decreased. The pandemic did not affect the working lives of the oldest people, which is understandable since the vast majority are retired and not active. Besides, as shown by a US study of older adults, this age group shows relatively high levels of resilience to COVID-19 pandemic stress and greater use of adaptive behavior [26]. Among respondents with the lowest primary education, one in five said their work life had changed for the better in a pandemic, while one in three said it had not changed. One in two people with higher education said the pandemic had not changed their lives. The youngest respondents best rated personal changes caused by COVID-19, and one in four said it worked out for them. A meta-analysis by Jones et al. indicated that adolescents’ response to the pandemic is inconclusive, although, globally, adolescents experience higher rates of anxiety [27]. For one in two people with vocational and secondary education, the pandemic had no impact on their personal lives. Researchers who analyzed the changes in personal and professional life caused by the COVID-19 pandemic among German-speaking workers during the first months of the pandemic reached similar conclusions, indicating that more than 40% of respondents saw no negative changes, and more than 10% saw positive impact from the pandemic [28].

Nevertheless, it is worth noting that among those declaring that the pandemic impacted their personal or professional lives, the majority said the impact was negative. It probably has to do with the fact that in a pandemic situation, the division between work and personal life is disrupted, which can translate into people’s psychological state, even if working at home was initially perceived positively by employees [29,30]. Most European Union workers surveyed indicate that working from home can lead to a deterioration of health and safety at work (61.6%) and affect increased stress levels (58.8%), which was also confirmed by studies among Polish workers [29,31].

In a pandemic, people cope with stress and anxiety very differently. Based on previous research on the acceptance of social situations after a pandemic, various coping strategies, such as behavioral activation and mindfulness practice, work well [32,33]. Nevertheless, situations of anxiety, anger, addiction, or depression may be a reason to seek specialized help in the form of pharmacology and/or advice from mental health professionals. As Tucci points out, psychologists, psychotherapists, and psychiatrists during a pandemic are essential to social coping with stress [34]. In our study, only one in seven respondents received counseling from a therapist/psychologist. In this group, only one in three respondents said the help of psychologists, and one in four for the help of psychiatrists, that the help was effective. Half of the respondents declared that they did not need the help of a therapist/psychologist due to the pandemic, just as 57.5% did not need the help of a psychiatrist. It is noteworthy that the treatment was successful in the small group of people who decided to take an antidepressant drug (13.1%), in most of them (92%).

Psychiatrists were slightly more likely to be visited by men than by women, with half of the psychiatrists’ patients being dissatisfied in the case of women. At the same time, dissatisfaction was very high in the case of men and involved most psychiatrist users. 

We showed a significant relationship between the use of psychological help due to pandemics and gender, age, education, and place of residence. Both women and men used psychological help equally, with men more often than women declaring that the support they received from a psychologist did not help them. Young people up to 34 years of age used psychological help most often, and the oldest people least often. In the senior group, it was only one in 12, and in the youngest group, up to 24, it was one in five. The youngest people still in high school—and therefore with only a primary education—used the help of specialists twice or even three times as often as older people and were three times more likely to rate this help positively. It is probably because psychological help in this age group is associated with a lower sense of taboo and stigma [35,36]. It has also been pointed out that young adults may be more likely to use psychological help because they are not only culturally accustomed to this kind of help but also because of the COVID-19 pandemic, which, especially in this group, caused a sense of restriction and discomfort. As young adults of all age groups are the most socially active, the pandemic significantly reduced this for them and—in the first phase—effectively prevented it [37]. The group of youngest respondents is also interesting in the context of pharmacology, as three-quarters of those with primary education declared that they did not need pharmacology because of the pandemic, on average several percentage points more than those with other levels of education.

The study also indicated a significant relationship between gender and income and the desire to return to life before the pandemic. As many researchers point out, nostalgia and hope play a significant role in coping with traumatic experiences. It may also explain how the surveyed Poles coped with the COVID-19 pandemic [38,39].

The impact of the COVID-19 pandemic on people’s mental health needs to be continuously monitored to develop social and health policy strategies because, as studies—including those reported in this article—show, the response to pandemic stress is determined by several social and demographic factors [40].

### Limitation of the Study

This study has some limitations. One of them is the study method which is closed within the framework of the quantitative method. At the same time, as we mentioned in the discussion, the topic of anxiety during the pandemic and coping with it would also require qualitative research. At the level of qualitative research, motivations regarding the use of specialized help, psychological and medical, as well as the lack of satisfaction with this help, should be investigated. Nevertheless, this is one of the first studies on this topic, on a representative sample of the entire population, one of the most populous European societies, and thus can provide a basis for further analysis and in-depth research.

## 5. Conclusions

The study pointed to several essential correlations in attitudes toward the pandemic, both from the social and public health perspectives. Presenting a slice of the public’s mental state precisely one year after the introduction of the pandemic, as well as the respondents’ self-reflections on the impact of the pandemic on their lives, juxtaposed with demographic categories, indicates the possible actions of adaptive mechanisms used by people, depending on gender, age, or education. 

The study observed a significant relationship between respondents’ self-assessment of their mental state and specialized help or pharmacology use. In all age categories, more people used psychological counseling, but it did not help them. On average, one in two people, regardless of education level, declared that they did not need the help of a psychologist or psychiatrist. The study regarded the ineffectiveness of the help of a therapist, psychiatrist, or drug treatment, as declared by most respondents. This observation requires a deeper diagnosis at the level of conducting further qualitative research to find out why respondents are so dissatisfied with the help of specialists. Patients reacting with maladaptation to a pandemic situation may require psychological rather than biological support; hence, psychological rather than pharmacological intervention should be recommended.

Our study supports the thesis that the effects of the pandemic, especially among young adults, should not be trivialized and that the impact of the COVID-19 pandemic on mental health and, in the long term, physical health requires a coordinated approach.

## Figures and Tables

**Figure 1 ijerph-20-01092-f001:**
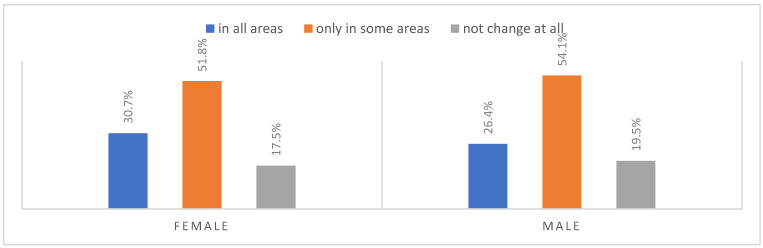
Scale of changes that the COVID-19 pandemic will cause by gender (N = 1000).

**Figure 2 ijerph-20-01092-f002:**
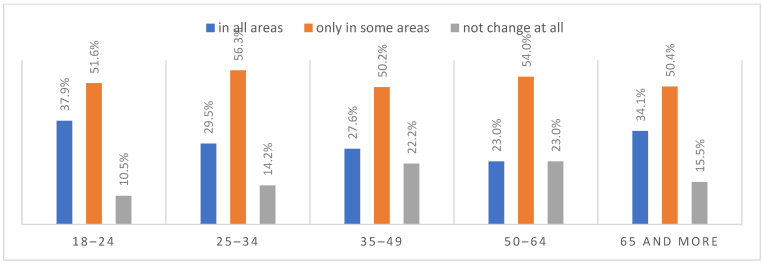
The magnitude of changes that the COVID-19 pandemic will cause by age (N = 1000).

**Figure 3 ijerph-20-01092-f003:**
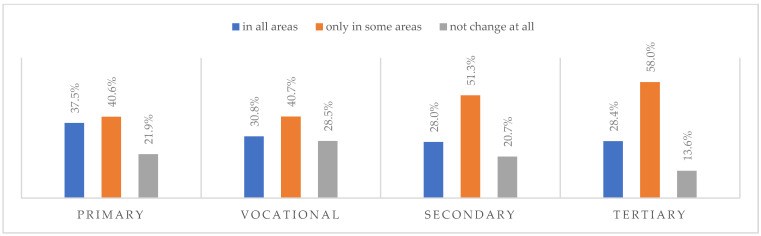
Scale of changes that the COVID-19 pandemic will cause by education (N = 1000).

**Figure 4 ijerph-20-01092-f004:**
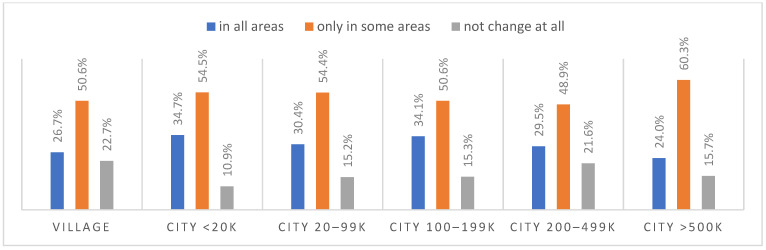
Scale of changes that the COVID-19 pandemic will cause, by place of residence (N = 1000).

**Figure 5 ijerph-20-01092-f005:**
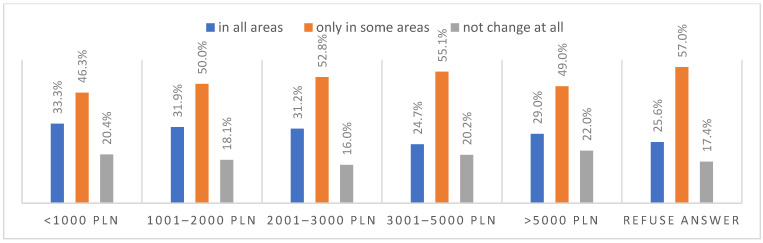
Scale of the changes that the COVID-19 pandemic will cause by income (N = 1000).

**Figure 6 ijerph-20-01092-f006:**
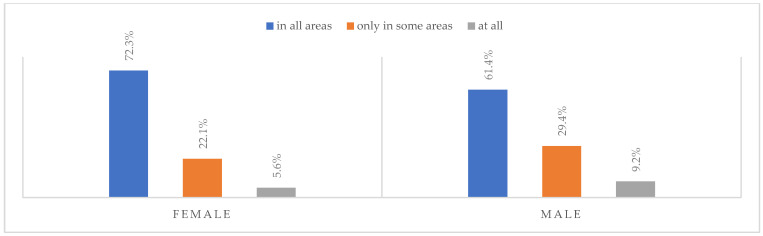
Willingness to return to life before the COVID-19 pandemic by gender (N = 1000).

**Figure 7 ijerph-20-01092-f007:**
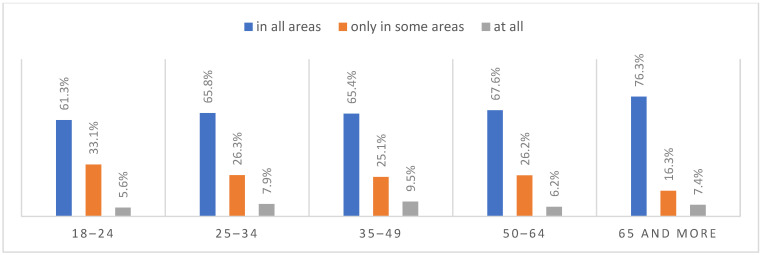
Willingness to return to life before the COVID-19 pandemic by age (N = 1000).

**Figure 8 ijerph-20-01092-f008:**
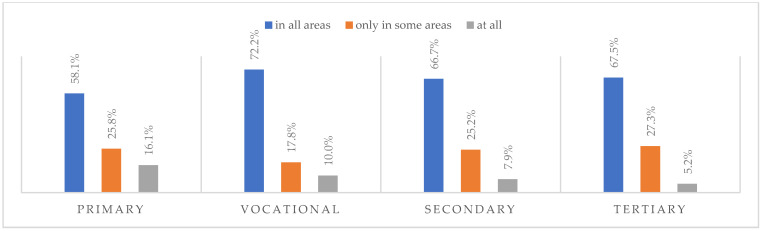
Willingness to return to life before the COVID-19 pandemic by education (N = 1000).

**Figure 9 ijerph-20-01092-f009:**
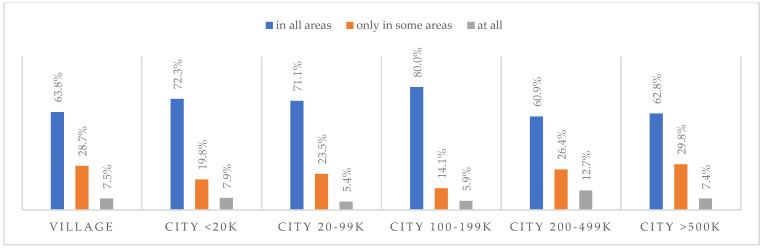
Willingness to return to life before the COVID-19 pandemic by place of residence (N = 1000).

**Figure 10 ijerph-20-01092-f010:**
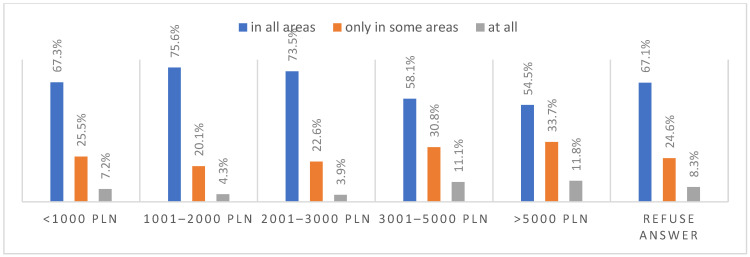
Willingness to return to life before the COVID-19 pandemic by income (N = 1000).

**Table 1 ijerph-20-01092-t001:** Sociodemographic characteristics of survey participants (N = 1000).

	N (%)
Total	1000 (100)
Gender	
Female	531 (53.1)
Male	469 (46.9)
Age (years)	
18–24	124 (12.4)
25–34	190 (19.0)
35–49	243 (24.3)
50–64	309 (30.9)
65 and more	135 (13.5)
Education	
Primary	32 (3.2)
Vocational	91 (9.1)
Secondary	460 (46.0)
Tertiary (subjects with university diplomas)	418 (41.8)
Place of residence	
Village (rural area)	401 (40.1)
Town to 20 k citizens	101 (10.1)
Town of 20–99 k citizens	205 (20.5)
City of 100–199 k citizens	85 (8.5)
City of 200–499 k citizens	87 (8.7)
City over 500 k citizens	121 (12.1)
Income (monthly)	
Up to 1000 PLN	55 (5.5)
1001–2000 PLN	209 (20.9)
2001–3000 PLN	231 (23.1)
3001–5000 PLN	18 (19.8)
Over 5000 PLN	100 (10.0)
Refusal to answer	207 (20.7)

**Table 2 ijerph-20-01092-t002:** Pandemic impact assessment by gender (N = 1000).

	N (%)		Test of Significance
	F	M	χ^2^/df/*p*-Value
The pandemic did not do people any good; it had rather negative effects.	334 (62.9)	284 (60.7)	0.518/1/0.474
The pandemic forced people to make positive changes in their lives; it had relatively positive effects.	197 (37.1)	184 (39.3)	

**Table 3 ijerph-20-01092-t003:** Pandemic impact assessment by age (N = 1000).

	N (%)					Test of Significance
	18–24 Years	25–34 Years	35–49 Years	50–64 Years	65 and More Years Old	χ^2^/df/*p*-Value
The pandemic did not do people any good; it had rather negative effects.	75 (60.5)	120 (63.5)	142 (58.4)	191 (61.8)	91 (67.4)	3.282/4/0.512
The pandemic forced people to make positive changes in their lives; it had relatively positive effects.	49 (39.5)	69 (36.5)	101 (41.6)	118 (38.2)	44 (32.6)	

**Table 4 ijerph-20-01092-t004:** Pandemic impact assessment by education (N = 1000).

	N (%)				Test of Significance
	Primary	Vocational	Secondary	Tertiary	χ^2^/df/*p*-Value
The pandemic did not do people any good; it had rather negative effects.	17 (53.1)	65 (71.4)	290 (63.0)	247 (59.1)	6.196/3/0.102
The pandemic forced people to make positive changes in their lives; it had relatively positive effects.	15 (46.9)	26 (28.6)	170 (37.0)	171 (40.9)	

**Table 5 ijerph-20-01092-t005:** Pandemic impact assessment by place of residence (N = 1000).

	N (%)						Test of Significance
	Village	Town up to 20 k	Town 20–99 k	City 100–199 k	City 200–499 k	City with Over 500 k	χ^2^/df/*p*-Value
The pandemic did not do people any good; it had rather negative effects.	247 (61.4)	62 (61.4)	130 (63.7)	53 (62.4)	56 (64.4)	72 (59.0)	0.997/5/0.963
The pandemic forced people to make positive changes in their lives; it had relatively positive effects.	155 (38.6)	39 (38.6)	74 (36.3)	32 (37.6)	31 (35.6)	50 (41.0)	

**Table 6 ijerph-20-01092-t006:** Pandemic impact assessment by income (N = 1000).

	N (%)						Test of Significance
	>1000 PLN	1001–2000 PLN	2001–3000 PLN	3001–5000 PLN	<5000 PLN	Data Denial	χ^2^/df/*p*-Value
The pandemic did not do people any good; it had rather negative effects.	35 (63.6)	138 (66.0)	156 (67.5)	109 (55.1)	57 (57.0)	123 (59.7)	10.060/5/0.074
The pandemic forced people to make positive changes in their lives; it had relatively positive effects.	20 (36.4)	71 (34.0)	75 (32.5)	89 (44.9)	43 (43.0)	83 (40.3)	

**Table 7 ijerph-20-01092-t007:** Assessment of the impact of the pandemic on the working lives of respondents by gender (N = 1000).

Q. Has the Pandemic Forced You to Change Your Work Life?	N (%)		Test of Significance
	F	M	χ^2^/df/*p*-Value
Yes, and it worked out for me	57 (10.7)	57 (12.2)	0.514/2/0.773
Yes, but it did not work out for me	165 (31.1)	143 (30.6)	
No, the pandemic did not affect it in any way	309 (58.2)	268 (57.3)	

**Table 8 ijerph-20-01092-t008:** Assessment of the impact of the pandemic on the working lives of respondents by age (N = 1000).

Q. Has the Pandemic Forced You to Change Your Work Life?	N (%)					Test of Significance
	18–24 Years	25–34 Years	35–49 Years	50–64 Years	65 and More Years Old	χ^2^/df/*p*-Value
Yes, and it worked out for me	19 (15.3)	27 (14.2)	33 (13.6)	28 (9.1)	7 (5.2)	38.553/8/<0.001
Yes, but it did not work out for me	50 (40.3)	60 (31.6)	80 (32.9)	97 (31.5)	22 (16.3)	
No, the pandemic did not affect it in any way	55 (44.4)	103 (54.2)	130 (53.5)	183 (59.4)	106 (78.5)	

**Table 9 ijerph-20-01092-t009:** Assessment of the impact of the pandemic on the working lives of respondents by education (N = 1000).

Q. Has the Pandemic Forced You to Change Your Work Life?	N (%)				Test of Significance
	Primary	Vocational	Secondary	Tertiary	χ^2^/df/*p*-Value
Yes, and it worked out for me	7 (21.9)	12 (13.2)	41 (8.9)	55 (13.2)	14.367/6/0.026
Yes, but it did not work out for me	5 (15.6)	24 (26.4)	137 (29.8)	142 (34.0)	
No, the pandemic did not affect it in any way	20 (62.5)	55 (60.4)	282 (61.3)	221 (52.9)	

**Table 10 ijerph-20-01092-t010:** Assessment of the impact of the pandemic on the working lives of respondents by place of residence (N = 1000).

Q. Has the Pandemic Forced You to Change Your Work Life?	N (%)						Test of Significance
	Village	Town up to 20 k	Town 20–99 k	City 100–199 k	City 200–499 k	City with Over 500 k	χ^2^/df/*p*-Value
Yes, and it worked out for me	50 (12.5)	9 (8.9)	22 (10.8)	8 (9.3)	5 (5.7)	21 (17.4)	11.814/10/0.298
Yes, but it did not work out for me	117 (29.2)	30 (29.7)	62 (30.4)	26 (30.2)	33 (37.9)	41 (33.9)	
No, the pandemic did not affect it in any way	234 (58.4)	62 (61.4)	120 (58.8)	52 (60.5)	49 (56.3)	59 (48.8)	

**Table 11 ijerph-20-01092-t011:** Assessment of the impact of the pandemic on the working lives of respondents by income (N = 1000).

Q. Has the Pandemic Forced You to Change Your Work life?	N (%)						Test of Significance
	>1000 PLN	1001–2000 PLN	2001–3000 PLN	3001–5000 PLN	<5000 PLN	Data Denial	χ^2^/df/*p*-Value
Yes, and it worked out for me	4 (7.4)	22 (10.5)	27 (11.7))	21 (10.6)	21 (21.0)	20 (9.7)	17.546/10/0.063
Yes, but it did not work out for me	21 (38.9)	54 (25.8)	70 (30.3)	70 (35.4)	31 (31.0)	62 (30.0)	
No, the pandemic did not affect it in any way	29 (53.7)	133 (63.6)	134 (58.0)	107 (54.0)	48 (48.0)	125 (60.4)	

**Table 12 ijerph-20-01092-t012:** Assessment of the impact of the pandemic on the personal lives of respondents by gender (N = 1000).

Q. Has the Pandemic Forced You to Change Your Personal Life?	N (%)		Test of Significance
	F	M	χ^2^/df/*p*-Value
Yes, and it worked out for me	52 (9.8)	62 (13.2)	3.101/2/0.212
Yes, but it did not work out for me	241 (45.3)	210 (44.8)	
No, the pandemic did not affect it in any way	239 (44.9)	197 (42.0)	

**Table 13 ijerph-20-01092-t013:** Assessment of the impact of the pandemic on the personal lives of respondents by age (N = 1000).

Q. Has the Pandemic Forced You to Change Your Personal Life?	N (%)					Test of Significance
	18–24 Years	25–34 Years	35–49 Years	50–64 Years	65 and More Years Old	χ^2^/df/*p*-Value
Yes, and it worked out for me	24 (19.4)	26 (13.7)	32 (13.2)	24 (7.8)	8 (6.0)	32.674/8/<0.001
Yes, but it did not work out for me	63 (50.8)	96 (50.5)	106 (43.8)	130 (42.1)	55 (41.0)	
No, the pandemic did not affect it in any way	37 (29.8)	68 (35.8)	104 (43.0)	155 (50.2)	71 (53.0)	

**Table 14 ijerph-20-01092-t014:** Assessment of the impact of the pandemic on the personal lives of respondents by education (N = 1000).

Q. Has the Pandemic Forced You to Change Your Personal Life?	N (%)				Test of Significance
	Primary	Vocational	Secondary	Tertiary	χ^2^/df/*p*-Value
Yes, and it worked out for me	8 (25.0)	10 (11.0)	44 (9.6)	52 (12.5)	14.804/6/0.022
Yes, but it did not work out for me	11 (34.4)	35 (38.5)	200 (43.6)	204 (48.9)	
No, the pandemic did not affect it in any way	13 (40.6)	46 (50.5)	215 (46.8)	161 (38.6)	

**Table 15 ijerph-20-01092-t015:** Assessment of the impact of the pandemic on the personal lives of respondents by place of residence (N = 1000).

Q. Has the Pandemic Forced You to Change Your Personal Life?	N (%)						Test of Significance
	Village	Town up to 20 k	Town 20–99 k	City 100–199 k	City 200–499 k	City with over 500 k	χ^2^/df/*p*-Value
Yes, and it worked out for me	52 (12.9)	11 (10.9)	18 (8.8)	10 (11.9)	6 (6.9)	17 (14.0)	14.883/10/0.136
Yes, but it did not work out for me	162 (40.3)	50 (49.5)	97 (47.5)	33 (39.3)	44 (50.6)	64 (52.9)	
No, the pandemic did not affect it in any way	188 (46.8)	40 (39.6)	89 (43.6)	41 (48.8)	37 (42.5)	40 (33.1)	

**Table 16 ijerph-20-01092-t016:** Assessment of the impact of the pandemic on the personal lives of respondents by income (N = 1000).

Q. Has the Pandemic Forced You to Change Your Personal Life?	N (%)						Test of Significance
	>1000 PLN	1001–2000 PLN	2001–3000 PLN	3001–5000 PLN	<5000 PLN	Data Denial	χ^2^/df/*p*-Value
Yes, and it worked out for me	5 (9.3)	20 (9.5)	28 (12.1)	17 (8.6)	24 (24.0)	21 (10.1)	25.738/10/0.004
Yes, but it did not work out for me	28 (51.9)	87 (41.4)	97 (42.0)	104 (52.5)	41 (41.0)	93 (44.7)	
No, the pandemic did not affect it in any way	21 (38.9)	103 (49.0)	106 (45.9)	77 (38.9)	35 (35.0)	94 (45.2)	

**Table 17 ijerph-20-01092-t017:** Overall assessment of life changes caused by the COVID-19 pandemic by demographic category (N = 1000).

						Test of Significance			Test of Normality of Distribution Kolmogorov-Smirnov		
	Mean	95% M	SE	MD	OR	χ^2^	df	*p*-Value	Stat.	df	Sign.
Gender											
Female	3.73	3.66–3.81	0.040	4.00	0.925	15.770	4	0.003	0.314	532	<0.001
Male	3.54	3.45–3.63	0.046	4.00	0.998				0.259	469	<0.001
Age											
18–24	3.52	3.33–3.70	0.093	4.00	1.032	38.315	16	0.001	0.283	124	<0.001
25–34	3.55	3.39–3.71	0.080	4.00	1.108				0.278	190	<0.001
35–49	3.61	3.48–3.73	0.062	4.00	0.961				0.278	243	<0.001
50–64	3.71	3.61–3.80	0.049	4.00	0.860				0.301	309	<0.001
65 and more	3.81	3.66–3.97	0.077	4.00	0.892				0.295	135	<0.001
Education											
Primary	3.20	2.72–3.67	0.232	3.05	1.305	29.000	12	0.004	0.216	32	<0.001
Vocational	3.74	3.54–3.94	0.102	4.00	0.975				0.258	91	<0.001
Secondary	3.65	3.56–3.74	0.044	4.00	0.945				0.295	460	<0.001
Tertiary	3.65	3.56–3.74	0.046	4.00	0.948				0.293	418	<0.001
Place of residence											
Village (rural area)	3.60	3.51–3.70	0.048	4.00	0.956	27.799	20	0.114	0.281	401	<0.001
Town to 20 k citizens	3.63	3.42–3.83	0.104	4.00	1.040				0.285	101	<0.001
Town of 20–99 k citizens	3.63	3.51–3.76	0.064	4.00	0.913				0.320	205	<0.001
City of 100–199 k citizens	3.92	3.73–4.10	0.092	4.00	0.848				0.346	85	<0.001
City of 200–499 k citizens	3.71	3.52–3.91	0.098	4.00	0.910				0.237	87	<0.001
City over 500 k citizens	3.57	3.38–3.77	0.100	4.00	1.102				0.259	121	<0.001
Income (monthly)											
Up to 1000 PLN	3.66	3.38–3.95	0.141	4.00	1.042	31.626	20	0.047	0.291	55	<0.001
1001–2000 PLN	3.68	3.55–3.81	0.066	4.00	0.948				0.281	209	<0.001
2001–3000 PLN	3.67	3.54–3.73	0.067	4.00	1.022				0.293	231	<0.001
3001–5000 PLN	3.59	3.46–3.72	0.066	4.00	0.935				0.291	198	<0.001
Over 5000 PLN	3.39	3.18–3.60	0.106	4.00	1.058				0.233	100	<0.001
Refusal to answer	3.75	3.63–3.86	0.059	4.00	0.855				0.314	207	<0.001

**Table 18 ijerph-20-01092-t018:** Using specialist consultations and taking mood-improving medications—distribution by gender (N = 1000).

	N (%)		
	F	M	χ^2^/df/*p*-Value
Due to the pandemic did you consulted a mental health specialist (psychologist)?			
Yes, and it helped me a lot	35 (6.6)	14 (3.0)	11.293/3/0.010
Yes, but it didn’t help me	33 (6.2)	41 (8.8)	
No, but I do not exclude that I will do this in the future	173 (32.6)	176 (37.6)	
No, I don’t need this	290 (54.6)	237 (50.6)	
Due to the pandemic did you consulted a mental health practitioner (psychiatrist)?			
Yes, and it helped me a lot	27 (5.1)	9 (1.9)	19.038/3/<0.001
Yes, but it didn’t help me	33 (6.2)	59 (12.6)	
No, but I do not exclude that I will do this in the future	154 (29.0)	143 (30.5)	
No, I don’t need this	317 (59.7)	258 (55.0)	
Due to the pandemic did you started taking medications to improve your mood, antidepressants?			
Yes, and it helped me a lot	31 (5.8)	20 (4.3)	1.873/3/0.599
Yes, but it didn’t help me	39 (7.3)	41 (8.7)	
No, but I do not exclude that I will do this in the future	137 (25.8)	124 (26.4)	
No, I don’t need this	325 (61.1)	284 (60.6)	

**Table 19 ijerph-20-01092-t019:** Using specialist consultations and taking mood medications—distribution by age (N = 1000).

	N (%)					
	<24 Years	25–34 Years	35–49 Years	50–64 Years	65 and More Years Old	χ^2^/df/*p*-Value
Due to the pandemic did you consulted a mental health specialist (psychologist)?						
Yes, and it helped me a lot	10 (8.1)	12 (6.3)	8 (3.3)	14 (4.5)	5 (3.7)	23.237/12/0.026
Yes, but it didn’t help me	15 (12.1)	18 (9.5)	19 (7.9)	16 (5.2)	6 (4.5)	
No, but I do not exclude that I will do this in the future	48 (38.7)	64 (33.7)	94 (38.8)	103 (33.3)	40 (29.9)	
No, I don’t need this	51 (41.1)	96 (50.5)	121 (50.0)	176 (57.0)	83 (61.9)	
Due to the pandemic did you have consulted a mental health practitioner (psychiatrist)?						
Yes, and it helped me a lot	5 (4.0)	10 (5.3)	4 (1.7)	13 (4.2)	5 (3.7)	16.888/12/0.154
Yes, but it didn’t help me	17 (13.7)	17 (8.9)	25 (10.3)	22 (7.1)	10 (7.5)	
No, but I do not exclude that I will do this in the future	37 (29.8)	56 (29.5)	86 (35.5)	84 (27.2)	34 (25.4)	
No, I don’t need this	65 (52.4)	107 (56.3)	127 (52.2)	190 (61.5)	85 (63.4)	
Due to the pandemic did you started taking medications to improve your mood, antidepressants?						
Yes, and it helped me a lot	5 (4.0)	10 (5.3)	10 (4.1)	20 (6.5)	6 (4.5)	8.833/12/0.717
Yes, but it didn’t help me	9 (7.3)	13 (6.9)	21 (8.6)	26 (8.4)	11 (8.2)	
No, but I do not exclude that I will do this in the future	31 (25.0)	56 (29.8)	73 (30.0)	67 (21.7)	33 (24.6)	
No, I don’t need this	79 (63.7)	109 (58.0)	139 (57.2)	196 (63.4)	84 (62.7)	

**Table 20 ijerph-20-01092-t020:** Using specialist consultations and taking medications to improve mood—distribution by education (N = 1000).

	N (%)				
	Primary	Vocational	Secondary	Tertiary	χ^2^/df/*p*-Value
Due to the pandemic did you consulted a mental health specialist (psychologist)?					
Yes, and it helped me a lot	5 (15.6)	5 (5.6)	18 (3.9)	21 (5.0)	21.211/9/0.012
Yes, but it didn’t help me	5 (15.6)	7 (7.8)	25 (5.4)	36 (8.6)	
No, but I do not exclude that I will do this in the future	5 (15.6)	25 (27.8)	173 (37.7)	146 (34.9)	
No, I don’t need this	17 (53.1)	53 (58.9)	243 (52.9)	215 (51.4)	
Due to the pandemic did you have consulted a mental health practitioner (psychiatrist)?					
Yes, and it helped me a lot	3 (9.4)	3 (3.3)	13 (2.8)	17 (4.1)	13.689/9/0.134
Yes, but it didn’t help me	7 (21.9)	9 (10.0)	39 (8.5)	37 (8.9)	
No, but I do not exclude that I will do this in the future	6 (18.8)	24 (26.7)	133 (28.9)	134 (32.1)	
No, I don’t need this	16 (50.0)	54 (60.0)	275 (59.8)	230 (55.0)	
Due to the pandemic did you started taking medications to improve your mood, antidepressants?					
Yes, and it helped me a lot	2 (6.5)	6 (6.7)	18 (3.9)	25 (6.0)	13.318/9/0.149
Yes, but it didn’t help me	3 (9.7)	5 (5.6)	38 (8.3)	33 (7.9)	
No, but I do not exclude that I will do this in the future	3 (9.7)	19 (21.1)	113 (24.6)	12 (29.9)	
No, I don’t need this	23 (74.2)	60 (66.7)	290 (63.2)	235 (56.2)	

**Table 21 ijerph-20-01092-t021:** Using specialist consultations and taking mood medications—distribution by place of residence (N = 1000).

	N (%)						
	Village	Town up to 20 k	Town 20–99 k	City 100–199 k	City 200–499 k	City with over 500 k	χ^2^/df/*p*-Value
Due to the pandemic did you consulted a mental health specialist (psychologist)?							
Yes, and it helped me a lot	19 (4.7)	10 (9.9)	8 (3.9)	5 (5.8)	1 (1.1)	7 (5.7)	26.038/15/0.038
Yes, but it didn’t help me	26 (6.5)	6 (5.9)	17 (8.3)	9 (10.5)	6 (6.9)	11 (9.0)	
No, but I do not exclude that I will do this in the future	151 (37.6)	37 (36.6)	66 (32.2)	15 (17.4)	38 (43.7)	43 (35.2)	
No, I don’t need this	206 (51.2)	48 (47.5)	114 (55.6)	57 (66.3)	42 (48.3)	61 (50.0)	
Due to the pandemic did you have consulted a mental health practitioner (psychiatrist)?							
Yes, and it helped me a lot	12 (3.0)	7 (6.9)	8 (3.9)	3 (3.5)	0 (0.0)	6 (5.0)	20.654/15/0.148
Yes, but it didn’t help me	39 (9.7)	9 (8.9)	17 (8.3)	8 (9.4)	6 (6.9)	14 (11.6)	
No, but I do not exclude that I will do this in the future	120 (29.9)	33 (32.7)	64 (31.2)	13 (15.3)	32 (36.8)	34 (28.1)	
No, I don’t need this	230 (57.4)	52 (51.5)	116 (56.6)	61 (71.8)	49 (56.3)	67 (55.4)	
Due to the pandemic did you started taking medications to improve your mood, antidepressants?							
Yes, and it helped me a lot	19 (4.7)	9 (8.9)	12 (5.9)	3 (3.5)	0 (0.0)	9 (7.4)	14.327/15/0.501
Yes, but it didn’t help me	31 (7.7)	11 (10.9)	17 (8.3)	6 (7.0)	8 (9.2)	7 (5.8)	
No, but I do not exclude that I will do this in the future	106 (26.4)	28 (27.7)	52 (25.5)	19 (22.1)	25 (28.7)	29 (24.0)	
No, I don’t need this	245 (61.1)	53 (52.5)	123 (60.3)	58 (67.4)	54 (62.1)	76 (62.8)	

**Table 22 ijerph-20-01092-t022:** Using specialist consultations and taking medications to improve mood—distribution by income (N = 1000).

	N (%)						
	>1000 PLN	1001–2000 PLN	2001–3000 PLN	3001–5000 PLN	<5000 PLN	Data Denial	χ^2^/df/*p*-Value
Due to the pandemic did you consulted a mental health specialist (psychologist)?							
Yes, and it helped me a lot	4 (7.3)	7 (3.3)	11 (4.8)	7 (3.6)	7 (6.9)	13 (6.3)	15.201/15/0.437
Yes, but it didn’t help me	5 (9.1)	13 (6.2)	18 (7.8)	11 (5.6)	12 (11.9)	15 (7.2)	
No, but I do not exclude that I will do this in the future	16 (29.1)	74 (35.4)	70 (30.3)	82 (41.6)	35 (34.7)	73 (35.3)	
No, I don’t need this	30 (54.5)	115 (55.0)	132 (57.1)	97 (49.2)	47 (46.5)	106 (51.2)	
Due to the pandemic did you have consulted a mental health practitioner (psychiatrist)?							
Yes, and it helped me a lot	4 (7.4)	5 (2.4)	12 (5.2)	3 (1.5)	4 (4.0)	8 (3.8)	22.778/15/0.089
Yes, but it didn’t help me	5 (9.3)	19 (9.0)	19 (8.2)	27 (13.6)	12 (12.0)	11 (5.3)	
No, but I do not exclude that I will do this in the future	15 (27.8)	64 (30.5)	58 (25.1)	63 (31.8)	36 (36.0)	61 (29.3)	
No, I don’t need this	30 (55.6)	122 (58.1)	142 (61.5)	105 (53.0)	48 (48.0)	128 (61.5)	
Due to the pandemic did you started taking medications to improve your mood, antidepressants?							
Yes, and it helped me a lot	5 (9.30	13 (6.2)	13 (5.6)	7 (3.5)	5 (5.0)	8 (3.9)	18.465/15/0.239
Yes, but it didn’t help me	4 (7.4)	22 (10.5)	18 (7.8)	16 (8.1)	9 (9.0)	11 (5.3)	
No, but I do not exclude that I will do this in the future	14 (25.9)	51 (24.4)	50 (21.6)	66 (33.3)	30 (30.0)	49 (23.7)	
No, I don’t need this	31 (57.4)	123 (58.9)	150 (64.9)	109 (55.1)	56 (56.0)	139 (67.1)	

**Table 23 ijerph-20-01092-t023:** Self-assessment of the current mental state vs. using specialist and pharmacological support (N = 1000).

	Definitely Optimistic	Rather Optimistic	Neither Optimistic nor Pessimistic	Rather Pessimistic	Definitely Pessimistic	*p*-Value
Due to the pandemic, did you consult a mental health specialist (psychologist)?						
Yes, and it helped me a lot	7 (0.7)	15 (1.5)	8 (0.8)	17 (1.7)	1 (0.1)	<0.001
Yes, but it didn’t help me	4 (0.4)	20 (2.0)	34 (3.4)	13 (1.3)	4 (0.4)	
No, but I do not exclude that I will do this in the future	4 (0.4)	89 (8.9)	151 (15.1)	85 (8.5)	20 (2.0)	
No, I don’t need this	20 (2.0)	140 (14.0)	261 (26.1)	80 (8.0)	26 (2.6)	
Due to the pandemic, did you consult a mental health practitioner (psychiatrist)?						
Yes, and it helped me a lot	3 (0.3)	14 (1.4)	7 (0.7)	12 (1.2)	0 (0.0)	<0.001
Yes, but it didn’t help me	7 (0.7)	30 (3.0)	37 (3.7)	15 (1.5)	3 (0.3)	
No, but I do not exclude that I will do this in the future	5 (0.5)	70 (7.0)	123 (12.3)	79 (7.9)	20 (2.0)	
No, I don’t need this	20 (2.0)	150 (15.0)	287 (28.7)	90 (9.0)	28 (2.8)	
Due to the pandemic, did you start taking medications to improve your mood, antidepressants?						
Yes, and it helped me a lot	4 (0.4)	20 (2.0)	12 (1.2)	13 (1.3)	2 (0.2)	0.004
Yes, but it didn’t help me	2 (0.2)	21 (2.1)	33 (3.3)	17 (1.7)	7 (0.7)	
No, but I do not exclude that I will do this in the future	7 (0.7)	70 (7.0)	104 (10.4)	67 (6.7)	12 (1.2)	
No, I don’t need this	22 (2.2)	154 (15.4)	304 (30.4)	99 (9.9)	30 (3.0)	

## Data Availability

The data presented in this study are available on request from the corresponding author.

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
