# Peer review of "The Global Pandemic as a Life-Changer? Medical, Psychological, or Self Help during COVID-19 Pandemic: A Cross-Sectional Representative Study"

_ijerph, 2023, doi:10.3390/ijerph20021092_

Round 1

Reviewer 1 Report

General comments

=============

This paper aims to discuss determine how the public in different social groups and age categories assessed the impact of the pandemic on their personal and professional lives. I would like to express several concerns and provide some comments and suggestions as follows. Hopefully, the following comments and suggestions will be helpful for improving this paper.

 =============

Major comments

---------------------

1. This paper still lacks further explanation of mental health of the public and creating patterns of public health policy action, especially the reference to the past research about the emotion behavior in pandemic.

2. For the literature, it is recommended that the literature related to compare studies from mental health of the public and creating patterns of public health policy action, in this study, and to indicate the results of studies in other research.  

 Minor comments

---------------------

3. This study used a random sampling method. And the questionnaire consisted of 20 questions. The authors should explain more about how to make them. Why choose these questions? How to make sure the samples are correct? For example, the samples are subjects with university diplomas 41.8%?

4.The authors can discuss results, implications for managers and scope for future researches in different sections to enhance readability

Author Response

Thank you very much for your careful review of our manuscript. All the comments provided were important and were very carefully considered. Below you will find responses to each comment.

General comments

=============

This paper aims to discuss determine how the public in different social groups and age categories assessed the impact of the pandemic on their personal and professional lives. I would like to express several concerns and provide some comments and suggestions as follows. Hopefully, the following comments and suggestions will be helpful for improving this paper.

 =============

Major comments

---------------------

  1. This paper still lacks further explanation of mental health of the public and creating patterns of public health policy action, especially the reference to the past research about the emotion behavior in pandemic.

Thank you for this indication. In the introduction, we referenced an important meta-analysis that addresses emotion-related behaviors in the context of pandemics and other publications that reference mental health in pandemic; Lines: 92-100

  1. For the literature, it is recommended that the literature related to compare studies from mental health of the public and creating patterns of public health policy action, in this study, and to indicate the results of studies in other research.  

Thank you for this comment. In the Discussion, we pointed out several solutions that can be used to create public health policy in the event of another pandemic. We supported our conclusions with additional sources. In addition, we have moved one of the essential conclusions of the study to the Conclusions.

 Minor comments

---------------------

  1. This study used a random sampling method. And the questionnaire consisted of 20 questions. The authors should explain more about how to make them. Why choose these questions? How to make sure the samples are correct? For example, the samples are subjects with university diplomas 41.8%?

The questionnaire consisted of 20 questions, 6 of which were demographic questions and 14 factual questions. Demographic questions included information on age, year of birth, gender, place of residence, education, and income. The factual questions were designed and validated (ad-hoc) by sociology, methodology, public health, psychology, and psychiatry experts. Based on the knowledge and experience of the experts, a questionnaire was developed to show as complete a picture as possible of the public's psychological well-being after the first year of the pandemic, within the constraints of the quantitative method and survey research. We added additional explanations in Lines: 131-137

How to make sure the samples are correct? For example, the samples are subjects with university diplomas 41.8%?

The sample for the survey was selected by a research company that, thanks to its international certifications, guarantees very high quality in the implementation of quantitative research. The high percentage of people with higher education is due to the limitations of random sampling, which does not guarantee a perfect representation of the amounts in each society. For this reason, random-quota sampling is a much better solution. Second, the number of people with higher education in Poland is high, at about 33% of people. Third, the education system in Poland is very complicated, which may ensure that a few percent of people declare that they have a university degree. However, they do not formally have one.

4.The authors can discuss results, implications for managers and scope for future researches in different sections to enhance readability

Thank you very much for this indication. As we prepare our following publication based on the research findings, we will undoubtedly devote more space to the implications of shaping public health policy during a pandemic.

Thank you once again!

Reviewer 2 Report

Dear Authors, 

The paper is interesting and deals with a topical issue. The aim was to test the changes in people’s life following the pandemic. 

In my opinion, some parts of the article need to be adjusted in order to further clarify your study. 

First of all, I suggest that information about measuring instruments should also be included in the abstract. 

In addition, I suggest that you better clarify the nature of the questionnaire you have created to collect the data, explaining the questions well (by adding examples), how to respond and how to administer (online, In presence, in group, individually?). 

I suggest to review also the font used for the writing of the article which is different between the one present in the introduction (I think times new Roman) and the one used for the tables and for the drafting of the discussions and conclusions. 

Finally, I found the description of the results and the part of the discussions well written and structured, full of all the necessary information and complete bibliography to support your study. 

I hope you will appreciate my suggestions in order to further enrich your interesting study.

Author Response

Thank you very much for your careful review of our manuscript. All the comments provided were important and were very carefully considered. Below you will find responses to each comment.

Dear Authors, 

The paper is interesting and deals with a topical issue. The aim was to test the changes in people’s life following the pandemic. 

In my opinion, some parts of the article need to be adjusted in order to further clarify your study. 

First of all, I suggest that information about measuring instruments should also be included in the abstract. 

Thank you very much. We have added this. Lines: 16-17

In addition, I suggest that you better clarify the nature of the questionnaire you have created to collect the data, explaining the questions well (by adding examples), how to respond and how to administer (online, In presence, in group, individually?). 

Thank you. We have supplemented this. The survey was conducted online using the CAWI technique, as we wrote about in the Study design and population, but we have added this information to the Abstract.

I suggest to review also the font used for the writing of the article which is different between the one present in the introduction (I think times new Roman) and the one used for the tables and for the drafting of the discussions and conclusions. 

Thank you. We have checked and corrected it.

Finally, I found the description of the results and the part of the discussions well written and structured, full of all the necessary information and complete bibliography to support your study. 

I hope you will appreciate my suggestions in order to further enrich your interesting study.

Thank you very much!

Reviewer 3 Report

The authors should ask for the help of a native English-speaking proofreader

because there are some linguistic mistakes that should be fixed. The title

needs further thought - shortened and more accurate. 

The Abstract in its sub-sections needs re-organization and it does not

adequately summarise the gist of the study.

Also, the writing style of the manuscript is not overall academic and formal.

The article is proposed to be supplemented with a flowchart illustrating the

research technique. A review of the literature is insufficient. It is critical to

include some recent work (2018–2020) in the literature review. A literature

review should be added in order to illustrate the central topic in a more

detailed way. Some further explanations and interpretations are required for

the results.

It is recommended to include a well-organized discussion of the

findings, strengths, and limitations of the present project with additional

explanation/details and a conclusion with future work.

I think the submission holds promise, but comprehensive editing is required.

Author Response

Thank you very much for your careful review of our manuscript. All the comments provided were important and were very carefully considered. Below you will find responses to each comment.

The authors should ask for the help of a native English-speaking proofreader

because there are some linguistic mistakes that should be fixed.

Thank you. Indeed, at this stage of the work, the manuscript had language errors, which we checked and tried to remove. The paper has been linguistically checked several times, including by a native speaker of English. We are confident that the minor linguistic errors will be corrected in the editorial editing process of the article.

The title needs further thought - shortened and more accurate. 

We realize that the title is long, but it fully corresponds to what is contained in the article, and we decided not to shorten it.

The Abstract in its sub-sections needs re-organization and it does not

adequately summarise the gist of the study.

Thank you for this comment. We have completed and corrected it.

Also, the writing style of the manuscript is not overall academic and formal.

Given the extensive academic and publication experience of the manuscript authors, we have taken every care to ensure that the article has formal and scientific overtones.

The article is proposed to be supplemented with a flowchart illustrating the

research technique.

The survey described here uses both simple random sampling and a standard, simple technique for selecting respondents for the survey. Delineating the CAWI technique will not bring value to the manuscript.

A review of the literature is insufficient. It is critical to

include some recent work (2018–2020) in the literature review. A literature

review should be added in order to illustrate the central topic in a more

detailed way. Some further explanations and interpretations are required for

the results.

It is recommended to include a well-organized discussion of the

findings, strengths, and limitations of the present project with additional

explanation/details and a conclusion with future work.

Thank you for this indication. Once again, we have reviewed the available literature on the research topic and included more than a dozen essential papers for Discussion, referring to them. We completed the Discussion and reformulated the structure. We separated the Limitations of the study and completed the Conclussions.

I think the submission holds promise, but comprehensive editing is required.

Thank you. We conducted a thorough review of our manuscript and made nearly a thousand corrections and additions that make the article more scientifically and linguistically valuable.

Thank you again!

Round 2

Reviewer 3 Report

The authors have implemented most of the suggested changes.

I think that it is adequate for publication.